# Orchid NAC Transcription Factors: A Focused Analysis of *CUPULIFORMIS* Genes

**DOI:** 10.3390/genes13122293

**Published:** 2022-12-05

**Authors:** Maria Carmen Valoroso, Francesca Lucibelli, Serena Aceto

**Affiliations:** 1Department of Agricultural Sciences, University of Napoli Federico II, 80055 Portici, Italy; 2Department of Biology, University of Naples Federico II, 80126 Napoli, Italy

**Keywords:** NAC transcription factors, *CUP* genes, flower development, flower symmetry, gene expression, Orchidaceae

## Abstract

Plant transcription factors are involved in different developmental pathways. NAC transcription factors (No Apical Meristem, *Arabidopsis thaliana* Activating Factor, Cup-shaped Cotyledon) act in various processes, e.g., plant organ formation, response to stress, and defense mechanisms. In *Antirrhinum majus,* the NAC transcription factor CUPULIFORMIS (CUP) plays a role in determining organ boundaries and lip formation, and the CUP homologs of *Arabidopsis* and *Petunia* are involved in flower organ formation. Orchidaceae is one of the most species-rich families of angiosperms, known for its extraordinary diversification of flower morphology. We conducted a transcriptome and genome-wide analysis of orchid *NAC*s, focusing on the No Apical Meristem (NAM) subfamily and *CUP* genes. To check whether the *CUP* homologs could be involved in the perianth formation of orchids, we performed an expression analysis on the flower organs of the orchid *Phalaenopsis aphrodite* at different developmental stages. The expression patterns of the *CUP* genes of *P. aphrodite* suggest their possible role in flower development and symmetry establishment. In addition, as observed in other species, the orchid *CUP1* and *CUP2* genes seem to be regulated by the microRNA, miR164. Our results represent a preliminary study of NAC transcription factors in orchids to understand the role of these genes during orchid flower formation.

## 1. Introduction

Plant transcription factors control many developmental processes, regulating the temporal and spatial expression of genes involved in plant growth and development. Among more than 2500 transcription factors present in plant genomes, a large number is involved in the development of the flower [1].

The *NAC* genes form one of the largest plant-specific families of transcription factors involved in different developmental and physiological processes such as organ formation [2,3,4], root development [5,6,7], organ boundary formation [8], plant defense [9,10,11], and response to abiotic stress [12,13,14].

Their name derives from the *NAM* gene (*No Apical Meristem*) in *Petunia*, *ATAF* (*A. thaliana Activating Factor*), and *CUC* (*Cup-shaped Cotyledon*) in *Arabidopsis*. The *Petunia nam* mutants lack a shoot apical meristem and show defects in organ boundaries [4]. This same phenotype characterizes the *Arabidopsis cuc1* and *cuc2* mutants and the *cupuliformis* (*cup)* mutant of the snapdragon, *A. majus* [2,8,15].

In the *A. thaliana* genome, more than 100 genes encode NAC transcription factors, divided into sixteen subfamilies based on amino acid sequence similarity [16,17]. The typical structure of a NAC protein has a conserved NAC domain at the N-terminus, with a twisted antiparallel β-sheet enclosed between two α-helices with five subdomains (A–E) [2,18]. The NAC domain, often found in two tandem repeats, contains positively charged amino acids possibly involved in DNA binding [19]. The NAC domain also enhances the protein homo- and hetero-dimerization, together with the C-terminal end [7,13,20]. This ability can positively or negatively regulate the activity of the NAC proteins.

The C-terminal region of the NAC proteins is poorly conserved; however, it includes a transcriptional activation region (TAR) containing serine, proline, and other simple amino acid repetitions [2,16]. Some atypical NAC transcription factors have a conserved NAC domain at the N-terminus and a conserved or absent C-terminal region [21].

Most studies into NAC transcription factors analyzed their function in root and leaf development and organ boundary formation [6,7,8,22,23,24,25]. However, in *A. majus,* the role of some NACs in the formation of the lip, the peculiar ventral petal of the snapdragon flower, was also explored. [15]. In this species, the *cup* mutant displays defects in the vegetative tissues, such as the *cuc* and *nam* mutants; it also shows an altered flower development. In particular, its corolla has an open mouth that lacks the folding characteristic of the palate and the lip regions. These mutant flowers lose bilateral symmetry, have similar petals, and their stamens are fused with the corolla tube [15,26]. CUP is a NAC transcription factor, and its involvement in flower formation was also described for *Arabidopsis* and *Petunia* [8,15,24].

The *CUP* gene of *A. majus* is differentially expressed during flower development. It is transcribed in the basal floral meristem region during the early developmental stages (5-10 days after floral meristem initiation, DAI). Later (11 to 14 DAI), it expands its expression to all the ventral and lateral parts of the flower [15]. During the early stages, the presence of the CUP protein at the organ boundaries is necessary to establish the organ tissue between confines. At the late stages of development, CUP may regulate the formation of the palate and lip in combination with other transcription factors encoded, for example, by the proximodistal genes *PALATE* and *LIP,* and the dorsoventral genes *DIVARICATA (DIV), CYCLOIDEA (CYC),* and *DICHOTOMA (DICH)* [8,15]. In addition to *A. majus*, the hypothetical role of the *CUP* gene in lip development was also proposed for two other Lamiales species, *Linaria maroccana* and *Mimulus guttatus* [15].

The activity of the *CUP* genes is regulated at the post-transcriptional level by the microRNA, miR164, which has a conserved target site on the transcripts of these genes [27,28,29]. In *Arabidopsis*, miR164 is encoded by three loci: *A, B,* and *C*. The isoforms *A* and *B* are identical, whereas the isoform *C* has a substitution of the canonical adenine with guanine at the last nucleotide position [30]. The isoforms miR164 *A* and *B* control stem growth, meristem differentiation, leaf shape, and senescence, while the isoform *C* is involved in floral organ formation [28,29,30,31].

To date, there are few or no studies into the NAC transcription factors in many flowering plant species. For this reason, we decided to analyze these transcription factors in Orchidaceae, one of the most species-rich families of angiosperms, focusing on the genes involved in flower organ formation in other species.

The bilaterally symmetric orchid flowers exhibit an extraordinary differentiation of morphology and adaptations [32,33]. Despite their great morphological diversity, the orchid flowers share a typical structure composed of three outer tepals, two laterals inner tepals, and a highly diversified median inner tepal (lip or labellum) [34]. Male and female reproductive tissues are fused to form the column, with the pollen grains at the apex and the ovary at the base [34].

As in all flowering plants, the genetic program at the basis of flower organ formation in orchids is realized mainly through the action of different MADS-box transcription factors [32,35,36,37,38,39,40,41,42,43]. In addition to this program, known as the ABCE model, an orchid-specific regulatory network involving the class B *MADS-box* genes *DEFICIENS*-like (*DEF*-like) and *AGAMOUS-LIKE 6* (*AGL6*) explains lip formation and symmetry determination [32,35,44].

However, the regulatory pathway underpinning the development of the orchid perianth organs is not restricted to the interaction of different classes of *MADS-box* genes. Some studies showed a conserved function of the MYB transcription factors DIV, RADIALIS (RAD), and DIV-and-RAD-Interacting-Factor (DRIF) in establishing bilateral symmetry of orchids [44,45] and the possible involvement of TCP and YABBY transcription factors in orchid lip formation [46,47]. Therefore, increasing evidence allows us to hypothesize that the molecular network driving orchid flower formation is more complex than previously thought.

Based on these premises and the role of the *CUP* genes in the flower organ formation of *Antirrhinum* and other species, we analyzed the orchid *CUP* genes to verify their possible involvement in orchid perianth formation.

We identified the orchid genes encoding NAC transcription factors and investigated their genomic organization, starting with the available orchid genomes and transcriptomes [35,42,43,48,49,50]. In addition, we analyzed the expression patterns of the *CUP* genes and miR164 in the orchid *P. aphrodite* at the early and late stages of flower development. *P. aphrodite* is an ornamental epiphytic orchid with many advantageous characteristics for cultivation and manipulation. Its inflorescence has many large flowers, and its genome was sequenced [48]. For these reasons, we used this species to conduct our expression analysis of the *CUP* genes and miR164.

## 2. Materials and Methods

### 2.1. Plant Material

The plants used in this study were grown in the Department of Biology at the University of Naples Federico II greenhouse (Napoli, Italy) under natural light and temperature. *P. aphrodite* (subfamily Epidendroideae) is commercially available, while *Orchis italica* (subfamily Orchidoideae) was collected in Sassano (Salerno, Italy) and is part of the Department of Biology orchid collection.

We sampled *P. aphrodite* flowers from three different plants at the B2 developmental stage (early, flower bud size 1–1.5 cm) and soon after anthesis (open flower, OF) (Figure 1). We dissected outer tepals, lateral inner tepals, lip, column, and ovary (not pollinated) from each *P. aphrodite* sample. In addition, we also collected leaf samples of *P. aphrodite* and *O. italica*. We stored the collected material in RNA-later (Ambion – Berlin, DE) until RNA extraction.

### 2.2. Identification, Phylogeny, and Conserved Motif Analysis of the Orchid NAC Genes

Using the conserved NAM domain (PFAM PF02365) as a query, we conducted a TBLASTN search against the available orchid genomes and transcriptomes deposited in the database Orchidstra 2.0 [51], OrchidBase [52], NCBI (https://www.ncbi.nlm.nih.gov/ accessed on 31 July 2022), and the transcriptome of *O. italica* [49]. When the orchid genome was available, we performed a genome search (*Apostasia shenzenica*, *Dendrobium catenatum*, *Phalaenopsis equestris*–Orchidbase; *P. aphrodite*–Orchidstra; *Vanilla planifolia*–NCBI). In the absence of a high-quality genome, we conducted a transcriptome search (*Cypripedium fomosanum*–Orchidbase; *O. italica*—reference transcriptome [49]).

We then virtually translated the identified orchid *NAC* nucleotide sequences to verify the presence of indels and/or stop codons within the coding sequence and evaluate their completeness. We discarded the sequences containing a partial NAM domain and/or stop codons.

In addition, we identified the *NAC* sequences of selected non-orchid species (*A. thaliana*, *A. majus*, *Amborella trichopoda*, *Asparagus officinalis*, *Zea mays*) using BLAST search and the Plaza 4.0 database [53]. The species name and accession number of the sequences used for the analyses are listed in Appendix A.

We aligned the amino acid sequences of the NAC proteins of *Arabidopsis* and each orchid species using ClustalW [54]. After manually adjusting the alignments, we constructed the Maximum Likelihood (ML) trees using MEGA7 [55] for each orchid species and *Arabidopsis*. We classified the orchid NAC sequences based on the ML trees and the subfamily classification of *Arabidopsis*. We then aligned the NAM amino acid sequences of orchid and non-orchid species and constructed the ML trees with 200 bootstrap replicates using MEGA7.

Using the orchid *NAC* coding sequences, we conducted a BLASTN search on the genomes of the orchids *D. catenatum* [43], *A. shenzenica* [42], *P. aphrodite* [48], *P. equestris* [35], and *V. planifolia* [50]. After reconstructing their genomic organization, we scanned the orchid genomic sequences of the *NAC* genes using CENSOR v4.2.22 software from Genetic Information Research Institute (GIRI), Cupertino, CA [56] to verify the presence of transposable/repetitive elements within introns.

Using the MEME v5.5 online tool (https://meme-suite.org/meme/) [57], we searched for shared amino acid motifs among the sequences of the orchid NAC proteins, setting the search parameters to a maximum number of 10 motifs and an optimum width from 6 to 50.

### 2.3. PCR Amplification and Sequencing of the CUP Genes of O. italica

To obtain information about the genomic organization of the highest number of orchid *CUP* genes, we included *O. italica* in our analysis. The genome is unavailable for this species; however, we have access to samples of *O. italica* that are part of the orchid collection of the Department of Biology (University of Naples Federico II). We extracted genomic DNA from leaves of *O. italica*, PCR-amplified and sequenced the *CUP* genes, thus reconstructing their exon/intron structure.

Starting with 500 mg of *O. italica* leaf material, we extracted the genomic DNA following the Doyle and Doyle protocol [58]. We designed specific primer pairs to PCR amplify the *CUP* genes of *O. italica* based on the nucleotide sequence of the transcript identified (Table 1). We conducted the amplification reactions in a final volume of 50 µL with the following reagent concentrations: 1X reaction buffer, 0.2 mM of each primer, 0.2 mM dNTP mix Platinum™ GC Enhancer, and 2 U *Taq* Hot-Start DNA Polymerase Invitrogen™ Platinum™ II (Invitrogen-Waltham, MA). The thermal cycle was the following: 94 °C for 2 min, 35 cycles of 94 °C for 15 sec, 60 °C for 15 sec, and 68 °C for a time ranging from 1:30 sec to 2 min, depending on the amplicon size.

We cloned the amplicons into the pSC-A-amp/kan vector (Agilent, Santa Clara, CA, USA), sequenced them using the T3 and T7 primers (Eurofins Genomics), and deposited the genomic sequences of the *CUP* genes of *O. italica* in GenBank with the accession numbers OP752215, OP752217, and OP752219. After reconstructing the genomic organization of the *O. italica CUP* genes, we verified the presence of transposable/repetitive elements within introns, as described earlier.

### 2.4. PCR Amplification and Sequencing of P. aphrodite CUP Transcripts 

We extracted total RNA from the floral buds of *P. aphrodite* at the B2 developmental stage using Trizol Reagent (Ambion – Berlin, DE) followed by DNase treatment. We reverse-transcribed 500 ng of total RNA using an Advantage RT-PCR kit (Clontech, Mountain View, CA, USA) with oligo dT and random primers.

We designed *CUP*-specific primer pairs starting with the sequences of *P. aphrodite* obtained from the in silico analysis (Table 1). We used 30 ng of first-strand cDNA to PCR amplify the transcripts of the *CUP* genes. We conducted each amplification reaction in a final volume of 50 µL, with the following reagent concentrations: 1X reaction buffer, 0.5 mM of each primer, and 2.5 U Wonder *Taq* DNA polymerase (Euroclone–Milan, IT). The thermal cycle was the following: 95 °C for 1 min, 35 cycles of 95 °C for 15 s, 57–59 °C for 15 s, and 72 °C for a time ranging from 30 s to 1:30 min, depending on the amplicon size.

We cloned and sequenced the amplicons as described before.

### 2.5. Expression Analysis

We extracted the total RNA from the outer and inner tepals, lips, columns, and ovaries collected from the *P*. *aphrodite* B2 stage and open flowers, and from the leaf tissue. We extracted, quantified and reverse-transcribed RNA as described in the previous paragraph.

We performed relative expression analysis of the *P. aphrodite CUP* genes by real-time qPCR on cDNA from flower organs and leaf tissue, using 18S as a reference gene [44,46]. We designed primer pairs specifically for the qPCR analysis (Table 1). We conducted the experiments in biological and technical triplicate using PowerUp™ SYBR™ Green Master Mix (Applied Biosystems™-Foster City, CA) following the manufacturer’s instructions and applying the following thermal conditions: 50 °C for 2 min, 95° for 2 min, 40 cycles of 95 °C for 15 s, 60 °C for 1 min.

We calculated the relative quantity (NRQ) ± SEM for each replicate normalized to the internal control gene [59]. After ANOVA analysis, we applied the Holm–Sidak post hoc test to assess the statistical significance of the differences in NRQ among the different tissues.

### 2.6. miR164 Analysis

We used the psRNATarget (2017 Update) online tool from The Zhao Lab^©^ Ardmore, OK [60] with stringent search parameters (maximum expectation 0.0) to search for the presence of the miR164 target site in the orchid *CUP* sequences.

To identify the eventual specific fragments produced by the cleavage of miR164 on the *P. aphrodite CUP* transcripts, we performed a modified 5′-RACE experiment. We designed specific reverse primers based on the position of the putative miRNA cleavage site in the *P. aphrodite CUP* transcripts (Table 1). We ligated the 5′-adaptor in the RLM-RACE GeneRace kit (Invitrogen-Waltham, MA) to the 5′-end of total RNA (500 ng) extracted from B2-stage buds of *P. aphrodite* without removing the 5′-cap. After reverse transcription, we amplified the cDNA using the reverse primers specific for *PaCUP1* and *PaCUP2* (Table 1) and the GeneRace 5′ forward primer.

We conducted a stem-loop real-time qPCR experiment to evaluate the expression pattern of the miR164 in *P. aphrodite*. We reverse-transcribed total RNA (150 ng) from floral organs of B2 floral buds, open flower (OF), and leaf tissue using a specific miR164 or 18S stem-loop primer (Table 1). We conducted the qPCR amplifications in biological and technical triplicate using the specific miR164 and 18S forward primers, and the universal stem-loop reverse primer. Amplification conditions, thermal cycle, NRQ calculation, and statistical analysis were performed as described in the previous paragraph.

### 2.7. Yeast Two-Hybrid Analysis

To check whether the proteins PaCUP1 and PaCUP2 of *P. aphrodite* interact with each other and form homodimers, we used the GAL4-based yeast two-hybrid (Y2H) system (Matchmaker two-hybrid system 3, Clontech-Mountain View, CA).

We amplified the full-length coding regions of the *PaCUP1*/*2* transcripts using specific primer pairs (Table 1). We cloned both the amplified coding regions into the pGADT7 (prey) and PGBKT7 (bait) expression vectors containing the activation (AD) and DNA-binding (BD) domains of the yeast transcription factor GAL4, respectively. We transformed the *Saccharomyces cerevisiae* strain AH109 [61] with all the combinations of pray and bait vectors, conducting each transformation in triplicate. After the yeast transformation, we verified the presence of the recombinant vectors by growing the yeast cells in Synthetic Defined (SD) medium lacking tryptophan and leucine in the presence of different concentrations (from 0 to 70 mM) of 3-aminotriazole (3AT). We transformed the empty vectors pGBKT7 or pGADT7 in combination with the recombinant vectors as negative controls.

In addition, we checked the autoactivation ability of the PaCUP1/2 proteins. To evaluate this, we grew the yeast transformed only with the recombinant PaCUP1 or PaCUP2 vector pGBKT7 (BD) on a selective medium lacking tryptophan and histidine in the presence of different concentrations (from 0 to 70 mM) of 3AT.

## 3. Results and Discussion

### 3.1. The NAC Transcription Factors of Orchids

Through BLAST analysis, we identified 423 orchid *NAC* transcripts. All sequences are listed in Appendix A. Almost all (370) have a complete coding sequence; partial sequences (53) containing the nucleotide region encoding for the NAC and TAR conserved motifs were included in the analysis. In addition, we downloaded the NAC protein sequences of the model species *A. thaliana* (111) and *A. majus* (77), the basal angiosperm *A. trichopoda* (42)*,* and the monocots *A. s officinalis* (95) and *Z. mays* (for this latter only the CUP proteins).

The number of *NAC* transcripts we identified in the different orchid species was variable, from 37 to 74. The species with the smallest number of sequences identified are *O. italica* (37) and *C. formosanum* (47). There is no assembled genome available for both species, so we used their transcriptomes to search the *NAC* sequences. The small number of *NACs* detected could be due to the incompleteness of their transcriptomes (e.g., obtained from specific tissues where some *NAC* genes are not expressed).

Based on sequence similarity, the NAC sequences can be classified into the subfamilies previously described in *Arabidopsis* [16] (Appendix A). As observed in other species, no orchid NAC belongs to the AtNAC3 and ANAC001 subfamilies, where only *Arabidopsis* sequences are included [16,62,63]. The analysis of the conserved domains of the NAC proteins reveals the presence of the five conserved subdomains (A-E) and the TAR regions specific to different subfamilies (Appendix A).

Scanning the available orchid genomes, we reconstructed the genomic organization of the orchid *NAC* genes. The intron number and size vary among the various orchid NAC subfamilies. Intron number ranges from one to ten, with the highest number found in the NAC2 subfamily; intron size ranges from 52 bp (intron 1 of the *A. shenzenica As014187* gene, SENU5 subfamily) to 20,287 bp (intron 1 of the *D. catenatum Dc021911* gene, ANAC063 subfamily).

Compared with *A. thaliana*, the *NAC* genes of orchids generally have a higher number of introns with a larger intron size in almost all cases (Appendix A). CENSOR analysis revealed traces of transposable/repetitive elements (e.g., DNA transposons, LTR, and Non-LTR Retrotransposons) within the introns of the orchid *NAC* genes displaying an intron number higher than *Arabidopsis* (Appendix A). The presence of repetitive elements and transposons in the *NAC* introns of orchids reflects a common feature of the orchid genomes. Indeed, transposable and repetitive elements can cover up to 60% of the orchid genomes [64], and orchid genes are often characterized by large introns containing repetitive and transposable elements, as observed in *KNOX* [65], *OitaAP2, OitaAG, OitaSTK* [66,67], and *MYB* genes [44].

The NAM subfamily of the *NAC* genes includes the highest number of sequences in almost all the species examined. Among them, the CUP transcription factors [68] are involved in the flower development of *A. majus* and other Lamiales [15]. For this reason, we focused our study on the NAM subfamily, in particular on the *CUP* genes.

Figure 2 shows the ML tree of the NAM transcription factors of orchids *A. thaliana*, *A. majus*, *A. officinalis,* and *Z. mays*. The tree topology shows a well-distinguished group that includes CUP-like proteins.

In addition to the NAC domain, NAM proteins have two transcriptional activation regions, TAR1 and TAR2 [16]. The Cf000662 protein in *C. formosanum* is the only orchid NAM protein containing both the TARs, whereas 34 orchid NAM proteins lack both regions (Appendix A), as observed in the *A. thaliana* AT1G76420, AT2G24430, and AT3G18400 sequences. The other orchid NAM sequences show only the TAR1 or TAR2 motif.

As in the other orchid NAC subfamilies, the NAMs have an intron size generally higher than *Arabidopsis*, as illustrated in Figure 3. Intron 2 shows the highest intron size variability. For example, in *D. catenatum*, *P. aphrodite*, *P. equestris,* and *V. planifolia,* the size of these introns ranges between 2001 and 9000 bp, values not observed in the *Arabidopsis* genes (Figure 3). In contrast, the coding region size is generally conserved between *Arabidopsis* and orchid species (Figure 3).

### 3.2. The Orchid CUP Genes: Phylogeny, Structure, and Expression Pattern

The *CUP* genes are involved in plant and flower development. In *Arabidopsis,* there are three *CUP* gene homologs [2,6]. *CUC1* and *CUC2* promote apical meristem growth during pre- and post-embryo development [5,31]. All three *Arabidopsis CUC* genes are involved in the maturation of flower reproductive organs; mutants of these genes reveal a phenotype strongly defective in the gynoecium and ovule structures [6,69]. In addition, the *CUC2* and *CUC3* genes regulate the formation of the border region of leaf primordia and their subsequent development [6,31]. In *A. majus,* the *CUP* gene is expressed, at an early stage of development, at a high level in the bud lateral–ventral region, regulating the boundary for lip development [15].

The ML tree obtained from the amino acid sequences of the orchid CUPs reveals the presence of two clades: the main includes the CUP1 and CUP2 proteins, and the other contains the CUP3 proteins (Figure 4a). This clade separation is not unique to orchids and is observed in other monocots, dicots, and early diverging angiosperm species [27,68,70,71]. The CUP1-2 clade is the most ancient, while duplication events occurring in angiosperms after divergence from gymnosperms generated the CUP3 clade [72].

The analysis of the conserved motifs reveals different patterns between the two clades (Figure 4b). Almost all the sequences have the characteristic NAM domain (motifs from 1 to 5) with few exceptions, possibly due to the incomplete CDS sequences isolated from the available transcriptomes. Motifs 6-7-8-10 are generally conserved within the CUP1-2 clade and absent in CUP3, whereas motif 9 is present only within the CUP3 clade.

Almost all the orchid *CUP* genes have three exons and two introns of variable size (Figure 4c). The *As001107* gene of the basal orchid *A. shenzenica* is the only orchid *CUP* gene with four exons and three introns.

To evaluate whether the *CUP* genes could be involved in orchid flower development, we conducted a preliminary analysis of their expression patterns in the early and late developmental stages of *P. aphrodite*.

The three *PaCUP* genes show different expression profiles in the perianth during development. At the early stage, the *PaCUP1-2* genes are mainly expressed in the lip, while *PaCUP3* is also highly expressed in the inner tepals (Figure 5a). After anthesis, the expression of the three *PaCUPs* decreases in the perianth (Figure 5a). At this stage, *PaCUP2* has higher expression in the inner tepals, while *PaCUP3* maintains high expression in the lip (Figure 5a).

All three *PaCUP* genes are highly expressed in the reproductive organs (column and ovary) at the early stage of development (Figure 5b). After anthesis, *PaCUP1* and *PaCUP3* decrease their expression level in the column and ovary, whereas the expression of *PaCUP2* increases (Figure 5b).

Only the *PaCUP3* gene is expressed in leaf tissue, as previously reported in *A. thaliana,* suggesting its possible role in the leaf [73] (Figure 5c).

At the early stage of flower development, the *P. aphrodite CUP* genes have an expression pattern similar to that of *A. majus* [15], being expressed in the lip and inner tepals. However, while in *A. majus* the *CUP* genes have a lateral–ventral profile, in the orchid flower the *CUP* expression is lateral–dorsal due to resupination. In most orchids, including *P. aphrodite*, resupination occurs before anthesis and is the 180° rotation of the pedicel and ovary, resulting in the reversal position of the dorsal and ventral organs in the mature flower. Consequently, although the lip is a dorsal structure, it has a ventral position [34].

As observed in *A. majus*, the expression pattern of the *CUP* genes in the orchid flower suggests their possible role in flower formation. The *CUP* genes might interact with other transcription factor encoding genes, e.g., *MYB* and *TCP*, involved in flower development and symmetry establishment [26,44,45,46,74,75,76]. Similar interactions were demonstrated in *A. majus*, where the CUPs can interact with the TCP proteins, suggesting their role in the boundary flower formation [8]. In addition, the phenotype of the *cup* mutant resembles that of the double mutant *cyc dich,* suggesting a functional correlation between *CUP* and *TCP* genes [8]. Furthermore, the expression of the *MYB* gene, *DIV,* could positively regulate the *CUP* expression in the labellum of *Antirrhinum* [15].

Recently, the orchid YABBY transcription factor, DL2, was proposed as a possible regulator of orchid lip formation [47]. In *P. equestris* there is a correlation between the expression pattern of *DL2* and the class-B *DEF*-like genes, *PeMADS2-PeMADS5*, responsible for orchid perianth formation [47]. Curiously, the expression pattern of *PaCUP1* in the lip is similar to that of *DL2* in *Phalaenopsis*. Both genes have high expression in the lip during the early developmental stages, followed by a decrease until the mature flower [47].

Our results suggest that, together with MADS-box and other transcription factors, the orchid *CUP* genes might be involved in the complex molecular pathway of orchid perianth formation.

### 3.3. Expression Pattern of miR164

The transcripts of the *CUP1-2* genes have a target site for the microRNA, miR164. This site, located within the region upstream of the 3′UTR, is absent in the *CUP3* lineage [71]. In *Arabidopsis,* microRNAs regulate the *CUP1-2* genes in many plant life processes. For example, the miR164c mutant has an altered number of petals, suggesting a role for miR164 in flower structure formation [77]. For these reasons, we investigated the presence of the miR164 target site on the orchid *CUP1-2* transcripts and compared the expression profile of this microRNA to that of the *PaCUP1-2* genes.

Using the psRNATarget online tool [60], we verified the presence of the putative miR164 target site on the *CUP* transcripts. Our results show that 12 orchid *CUP1-2* transcripts have the miR164 target site (Figure 6). As expected, no *CUP3* transcript had the miR164 target site.

We conducted the modified 5′-RACE experiment on the *PaCUP* transcripts of *P. aphrodite* in the tissues examined and did not detect any miR164 cleavage fragment of the expected size. The apparent absence of the cleavage product in the selected tissues may be due to the high instability of the mRNA fragment produced after cleavage by miR164.

To compare the expression patterns of miR164 and *PaCUP* genes, we conducted stem-loop qPCR experiments of miR164 on floral tissues from the B2 and OF stages of *P. aphrodite* (Figure 7a,b).

In the perianth, miR164 shows an expression trend opposite to that of the *PaCUP1-2* genes. At the B2 stage, the miR164 transcripts are almost undetectable, whereas they are expressed in the floral tissues after anthesis (Figure 7a), suggesting that the expression of miR164 reduces the number of the *PaCUP1-2* transcripts. These results are in agreement with those obtained in *A. majus,* where at the first stage of development, the *CUP1-2* genes regulate the boundary formation and are expressed in all flower primordia. Their expression then decreases before anthesis, limiting the presence of the *CUP* transcripts to specific parts of the flower [15].

In the reproductive structures of *P. aphrodite*, miR164 is not expressed at the B2 stage, allowing a high expression of the *PaCUP* genes. Later, at the OF stage, mir164 is expressed; however, only the transcript levels of *PaCUP1* decrease, in particular in the ovary, whereas *PaCUP2* is still highly expressed, suggesting that after anthesis, the levels of *PaCUP2* transcripts are not regulated by miR164. The activity of miR164 on the transcript levels of the homologs of the *CUP* genes was also proposed for *A. thaliana*, where miR164 is involved in reproductive structure formation and possibly modulates the transcript levels of the *CUC1-2* genes (homologs of the CUPs) involved in ovary and column formation [31,69].

Finally, miR164 is expressed in the leaf tissue of *P. aphrodite*, where the *PaCUP1-2* transcript levels are low. The miR164 of *A. thaliana* regulates the *CUC1-2* genes during leaf shape formation and senescence. Mutants of miR164 have an unusual leaf shape, with leaves showing a more pronounced dentition than the wild type [78]. Our results suggest that, also in orchids, miR164 seems to regulate the *PaCUP1-2* expression in the leaf.

### 3.4. PaCUP1–PaCUP2 Protein Interaction

The NAC transcription factors of *Arabidopsis* can form homo- and heterodimers due to their ability to interact through the NAC domains [25]. In addition, CUC1 and CUC2 in *Arabidopsis* can activate transcription due to the presence of acidic residues within the TAR domain [25]. In a Y2H assay, they activated the transcription of the yeast reporter genes [79]. For these reasons, we checked the autoactivation ability of the *P. aphrodite* PaCUP1-2 proteins and their ability to form homo- and heterodimers in the Y2H assay.

The yeast clones double-transformed with the recombinant vector pGBKT7 containing the full-length transcripts *PaCUP1* or *PaCUP2,* and the empty vector PGADT7. The clones were grown on a selective medium without tryptophan, lysine, or histidine in the presence of different concentrations of 3AT (from 0 to 70 mM). Only the PaCUP1 protein alone activated the transcription of the yeast *GAL4* gene in the absence of histidine and in the presence of up to 20 mM 3AT (Figure 8a). For this reason, the protein–protein interaction experiments were conducted in the presence of 3AT from 50 to 70 mM.

Our results reveal that PaCUP1 forms homodimers (Figure 8a), as observed in *Arabidopsis* [79], and, as expected, PaCUP1 and PaCUP2 proteins directly interact with each other (Figure 8b), showing that the CUP1-2 proteins have a conserved ability to cross-interact.

## 4. Conclusions

Orchidaceae represent approximately 10% of flowering plants, with many unique morphological, ecological, and physiological characteristics. Their cultivation and commercialization are a significant part of the worldwide floriculture trade. In this scenario, studies on the molecular basis of orchid flower development are particularly relevant, as they enrich the knowledge of how to develop specific breeding programs to obtain even more new cultivars and varieties.

For the first time in orchids, we analyzed the NAC transcription factor family, focusing on the *CUP* genes.

Our study reveals that orchid NAC proteins have a conserved domain organization and can be easily classified into different subfamilies based on their sequence.

The expression patterns of the *P. aphrodite CUP* genes suggest their possible involvement in flower formation and symmetry establishment. In addition, as previously described for different species, the transcript levels of orchid *CUP1* and *CUP2* genes seem to be regulated by miR164.

The data obtained in this work are preliminary and represent a starting point for understanding the role of these genes. Further investigations, in particular functional analyses, are needed to clarify the role of *CUP* genes during orchid flower formation.

## Figures and Tables

**Figure 1 genes-13-02293-f001:**
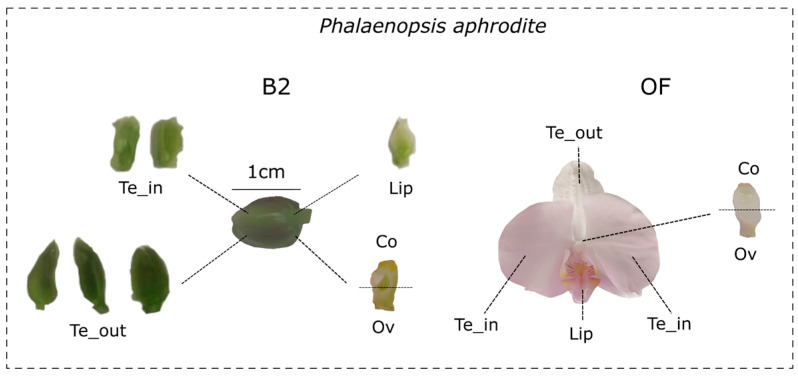
Flower and dissected organs of *P. aphrodite* at the B2 developmental stage (flower bud size 1-1.5 cm) and soon after anthesis (OF). Te_out, outer tepals; Te_in, lateral inner tepals; Co, column; Ov, ovary.

**Figure 2 genes-13-02293-f002:**
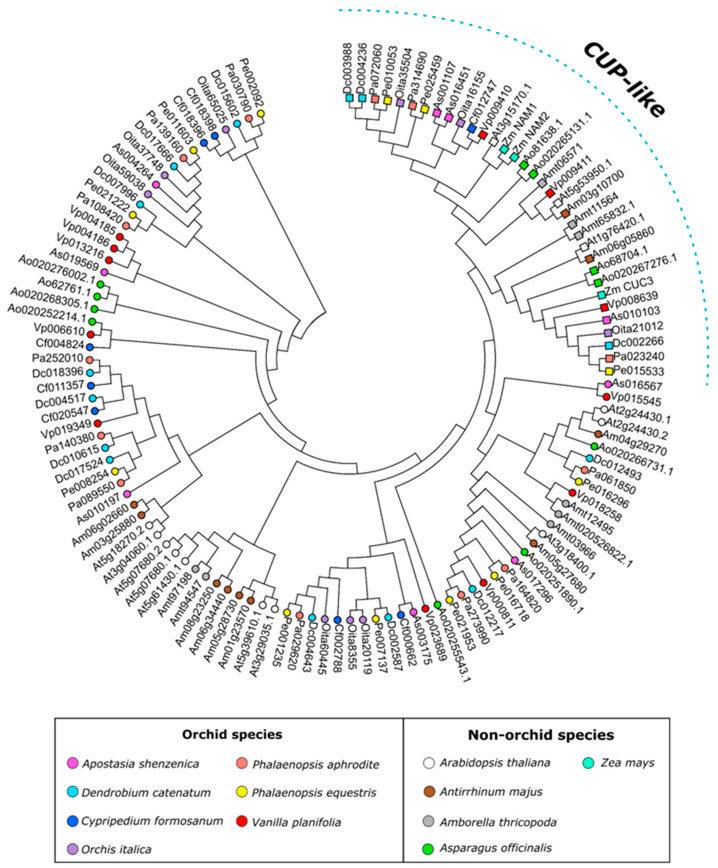
ML tree of the NAM subfamily. The sequences of the CUP and CUP-like proteins are marked with colored squares. The tree includes sequences from all the orchid and non-orchid species examined in this work.

**Figure 3 genes-13-02293-f003:**
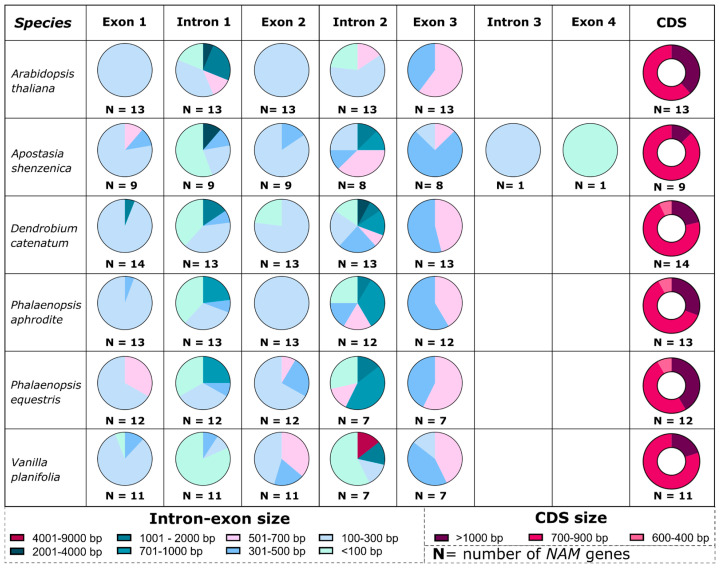
Comparison of *NAM* gene intron and exon sizes between *Arabidopsis* and orchids. For each species, the rows represent the intron/exon structure of the *NAM* genes. The different colors indicate the intron/exon size range. The pie charts show the percentage of genes with a specific size range for each intron/exon (see Appendix A). The last column shows the size variability in *NAM* gene CDSs.

**Figure 4 genes-13-02293-f004:**
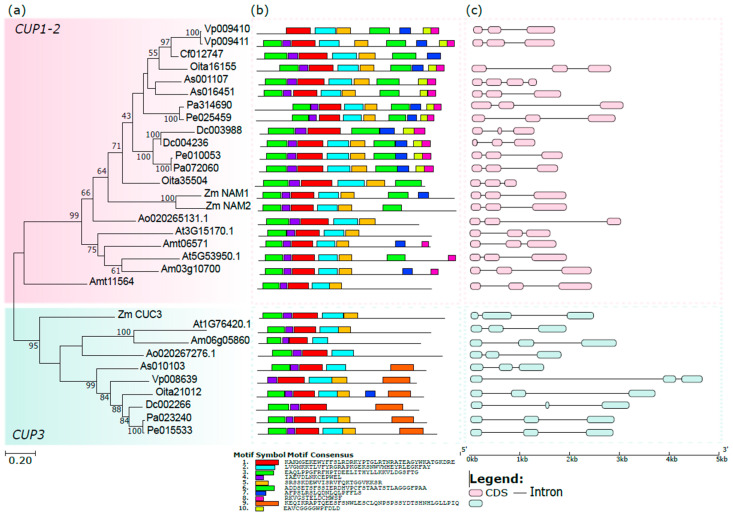
Phylogeny of orchid CUP proteins. (**a**) ML tree of the orchid CUP proteins, *A. thaliana*, *A. majus*, *A. trichopoda*, *A. officinalis,* and *Z. mays*; (**b**) conserved motifs of CUP proteins (see Appendix A); (**c**) genomic organization of orchid *CUP* genes and other species with available genomes.

**Figure 5 genes-13-02293-f005:**
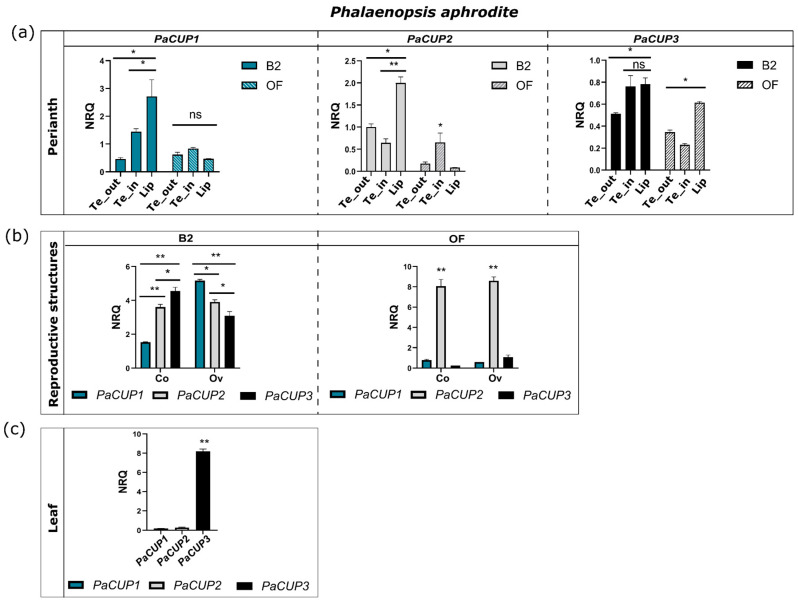
Expression patterns in *P. aphrodite PaCUP* genes. (**a**) perianth tissue; (**b**) reproductive tissue; (**c**) leaf. The expression level is reported as normalized relative quantity (NRQ) (reference gene 18S). The bars represent the SEM of the biological and technical replicates. The asterisks indicate statistically significant differences in the expression level at the B2 (early) and OF (late) stages. *p* values * <0.0004, ** <0.0001; ns, not significant. All the results are reported in Appendix A. Te_out, outer tepals; Te_in, inner tepals; Co, column; Ov, ovary.

**Figure 6 genes-13-02293-f006:**
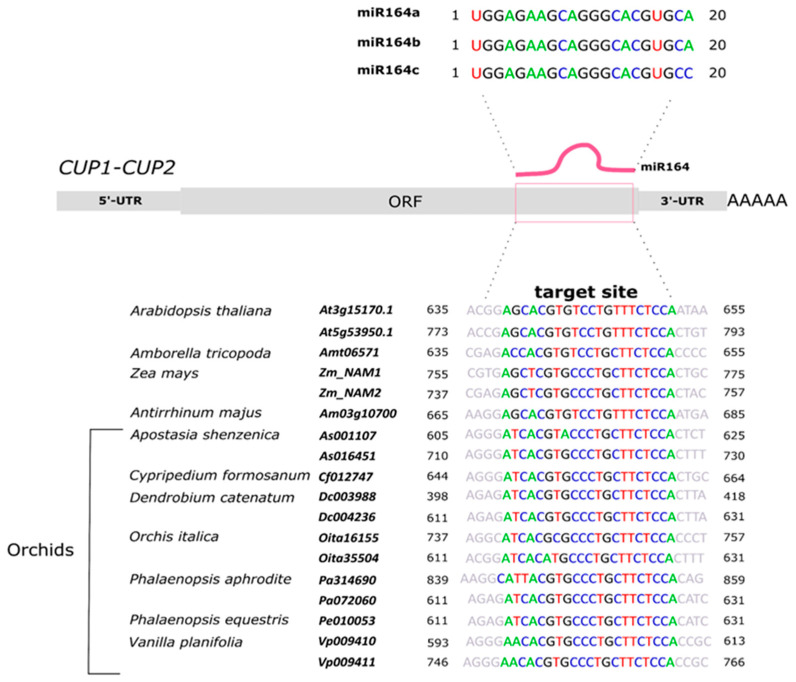
The miR164 target site on the *CUP* transcripts of orchid and non-orchid species.

**Figure 7 genes-13-02293-f007:**
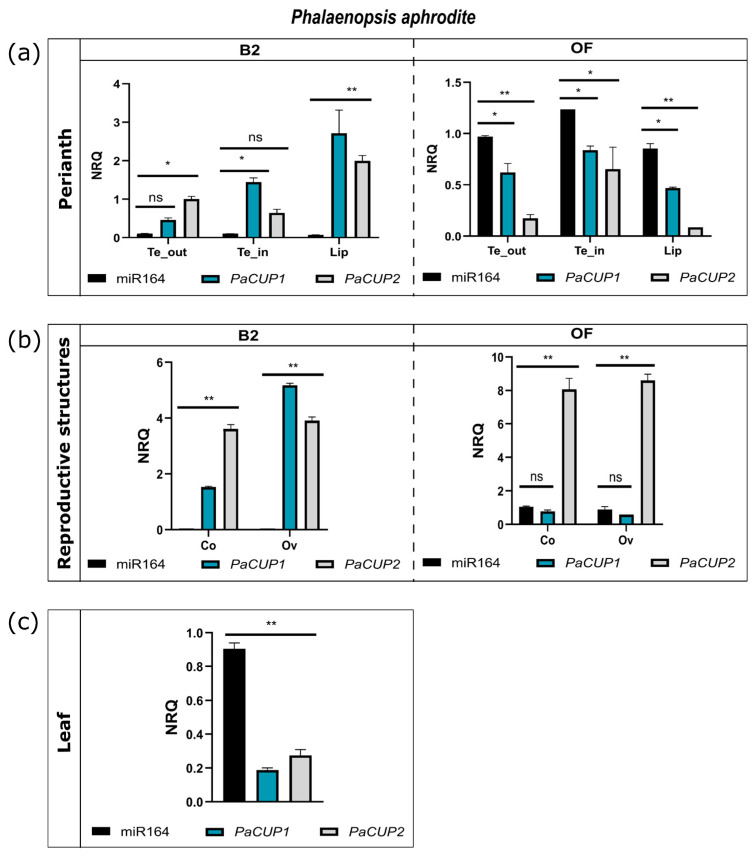
Expression patterns of miR164 in *P. aphrodite*. Comparison of the expression patterns of miR164 and *PaCUP1-2* in perianth structures (**a**) and reproductive organs (**b**) of *P. aphrodite* at the B2 and OF developmental stages. (**c**) Comparison of the expression pattern of miR164 and *PaCUP1-2* in a *P. Aphrodite* leaf. The expression level is reported as normalized relative quantity (NRQ). The bars represent the SEM of the biological and technical replicates. The asterisks indicate the statistically significant differences between the miR164 and *CUP1-2* NRQ values. *p* values * < 0.0257, ** < 0.0001. The results are reported in Supplementary Data S4. Te_out, outer tepals; Te_in, inner tepals; Co, column; Ov, ovary.

**Figure 8 genes-13-02293-f008:**
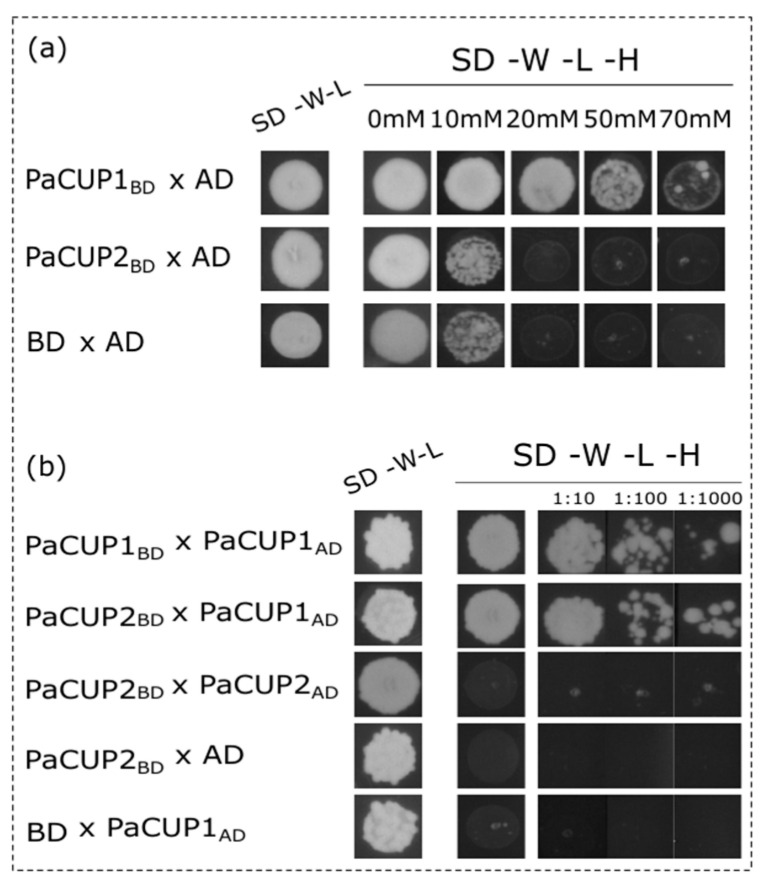
Yeast two-hybrid analysis to verify the interaction between the *P. aphrodite* PaCUP1 and PaCUP2 proteins. To check for the presence of plasmids, after double transformations, yeasts were grown in absence of tryptophan and leucine (SD-W-L medium); to verify the protein interactions, yeasts were grown in a medium lacking tryptophan, leucine, and histidine (SD-W-L-H medium). (**a**) Auto-activation experiments using the recombinant vector pGBKT7 containing the coding sequence of *PaCUP1* or *PaCUP2* and the empty vector pGADT7. Double transformation with the pGADT7 and pGBKT7 empty vectors are the negative control. (**b**) Interaction of the PaCUP1 and PaCUP2 proteins. 1:10, 1:100, and 1:1000 were the dilution factors applied to the yeast inoculation. AD, GAL4 activation domain; BD, GAL4 DNA-binding domain.

**Table 1 genes-13-02293-t001:** List of the primer sequences used for genomic (gDNA) and cDNA sequence isolation, qPCR analysis, miRNA and Yeast two-hybrid analysis.

Application	Gene	Forward (5’-3’)	Reverse (5’-3’)
**gDNA**	*Oita16155*	TCCTCCTCCTGACAACCCCT	GCTGTAGATCTCCCGGTCTT
	*Oita16155*	ATGAGTACAGGTTGGAGGGC	GCACCGCTTTCCTTCATCTTC
	*Oita35504*	CCCTCACTCTTCTGGCATGG	GGCATGTGATCCGTCTCGAT
	*Oita21012*	ACGTCACCTGTCTTGTTCGG	TTGCGGTCTCTTAGGCTGAA
	*Oita21012*	AGACTTCTCCTTCCGACACG	AGGTGTCTGCAGTTTGGTGG
	*Oita21012*		GCTGTAGATCTCCCGGTCTT
**cDNA**	*Pa_CUP1*	ACAAATCCCATACGCTTCGC	AACATGCAGTCCAACTCCGT
	*Pa_CUP2*	CTTCTAACTATCCCGGCCCC	ATCTCACACTCTGTCACCGG
	*Pa_CUP3*	GGCTTGAATCCTGCTTACAA	GGTGCTCCTCCTTGGATTGG
**qPCR**	*Pa_CUP1*	ACAAATCCCATACGCTTCGC	AACATGCAGTCCAACTCCGT
	*Pa_CUP2*	CTTCTAACTATCCCGGCCCC	ATCTCACACTCTGTCACCGG
	*Pa_CUP3*	GGCTTGAATCCTGCTTACAA	GGTGCTCCTCCTTGGATTGG
	*Pa18S*	TTAGGCCACGGAAGTTTGAG	ACACTTCACCGGACCATTCAA
**miRNA**	*miRNA164_Stloop*	GCGGCGTGGAGAAGCAGG	GTCGTATCCAGTGCAGGGTCCGAGGTATTCGCACTGGATACGACCGCACGTGC
	*Stloop_18S*		GTCGTATCCAGTGCAGGGTCCGAGGTATTCGCACTGGATACGACCACTTC
	*Stloop_UNIV*		GTGCAGGGTCCGAGGT
**Y2H**	*Pa_CUP1*	AGGATCCAAATGGAGAACTTCAGCCACCAT	AACTGCAGGATGATGTCATCCACTTAAAATCGT
	*Pa_CUP2*	AGGATCCAAATGGAGAACTTCAGTCAGCATTTCGAC	AACTGCAGTGGAAGCACAAAACATAATTAGCTC

## Data Availability

The data presented in this study are openly available in NCBI (https://www.ncbi.nlm.nih.gov/) with the accession numbers listed in Appendix A. The databases used in this work are the following: https://orchidstra2.abrc.sinica.edu.tw/orchidstra2/index.php; http://orchidbase.itps.ncku.edu.tw/EST/home2012.aspx; http://bioinfo.sibs.ac.cn/Am/; https://bioinformatics.psb.ugent.be/plaza/versions/plaza_v5_monocots/; https://bioinformatics.psb.ugent.be/plaza/versions/plaza_v5_dicots/.

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
