# Peer review of "Orchid NAC Transcription Factors: A Focused Analysis of CUPULIFORMIS Genes"

_genes, 2022, doi:10.3390/genes13122293_

Round 1

Reviewer 1 Report

This manuscript is about NAC genes of orchid. NAC is a big gene family with more than 100 members in higher plants, known to function in multiple biological process. This study identified NAC sequences from orchid genome by conventional bioinformatic approaches, cloned transcript sequence of cup gene from orchid P.aphrodite, profiled expression pattern of cup gene and miR164, tried to confirm that CUP was the target of miR164. Yeast two hybrid analysis is done to analyze protein interaction of CUP protein. The manuscript is well organized and impressively informative.

Major comments:

(1)    Section 2.1

This study used two species O. italica and P. Aphrodite. What is the reason selecting these two species? Because their flowers are totally different and the authors want to compare between them? Will the authors please give more explanation.

P3, L132-133, what is the orchid species of database Orchidstra and ORchidbase?

(2)    P3, l130-L135

They identify NAC genes in two orchid genomes and transcriptome of O.italica. Does that mean the final NAC genes is a union of set? How to treat those sequence with high identity (ortholog) between different database? Take them as the same sequence? Different sequences? It seems strange and complicated. Otherwise, show more detailed methodology.

(3)     Figure 3, figure 4

Gene structure off course must be very different between Arabidopsis and orchid. I cannot see the significance of this work. Anyhow, it is reasonable to compare between relative species.

(4)    Figure 5 and figure 7

Why not combine them?

(5)    Section 3.4

I see CUP1 interact with CUP2. There is no background why they study their interaction. Is this their initial purpose?

Small suggestions

(1)    Figure 2, figure legend

Please explain color by color, not by word.

(2)    P10, L354

“P values*<0,0004”. What is the inner reference gene?

(3)    P11, L398

There is figure legend of figure 6, but no figure.

Author Response

Dear Editor,

We thank you and the three reviewers for the positive feedback on our MS. We appreciate the comments and suggestions that helped us to improve our work.

The point-by-point answers to reviewers are listed below.

We hope that the revised version of the MS is now suitable for publication in Genes.

Best regards,

Maria Carmen Valoroso,

Francesca Lucibelli

Serena Aceto

REVIEWER 1

This manuscript is about NAC genes of orchid. NAC is a big gene family with more than 100 members in higher plants, known to function in multiple biological process. This study identified NAC sequences from orchid genome by conventional bioinformatic approaches, cloned transcript sequence of cup gene from orchid P.aphrodite, profiled expression pattern of cup gene and miR164, tried to confirm that CUP was the target of miR164. Yeast two hybrid analysis is done to analyze protein interaction of CUP protein. The manuscript is well organized and impressively informative.

Major comments:

(1)    Section 2.1

This study used two species O. italica and P. Aphrodite. What is the reason selecting these two species? Because their flowers are totally different and the authors want to compare between them? Will the authors please give more explanation.

ANSWER: To obtain information about the genomic organization of the highest number of orchid CUP genes,  we included Orchis italica in our analysis. The genome is unavailable for this species; however, we have access to samples of O. italica that are part of the orchid collection of the Department of Biology (University of Naples Federico II). We extracted genomic DNA from leaves of O. italica, PCR amplified and sequenced the CUP genes, thus reconstructing their exon/intron structure. We added a sentence about this in the Materials and Methods section. In addition, to avoid confusion, we modified Figure 1 including only the flower of P. aphrodite.

P3, L132-133, what is the orchid species of database Orchidstra and ORchidbase?

ANSWER: We added this information in the Materials and Method section.

(2)    P3, l130-L135

They identify NAC genes in two orchid genomes and transcriptome of O.italica. Does that mean the final NAC genes is a union of set? How to treat those sequence with high identity (ortholog) between different database? Take them as the same sequence? Different sequences? It seems strange and complicated. Otherwise, show more detailed methodology.

ANSWER: We modified the Methods section, clarifying the methodology applied to identify the orchid NACs. When the genome was available, we started with a genome search (Orchidstra/Orchidbase) and then checked the CDS on the transcriptome databases and NCBI. In the absence of a released genome, we conducted a transcriptome search. In addition, for Orchis italica, the CUP genes were PCR amplified and sequenced starting from the genomic DNA extracted from leaves.

(3)     Figure 3, figure 4

Gene structure off course must be very different between Arabidopsis and orchid. I cannot see the significance of this work. Anyhow, it is reasonable to compare between relative species.

ANSWER: Between distantly related species, the structure of specific genes can be conserved or divergent. In both Figures, we included Arabidopsis as model species where the NAM gene structure is well known. 

(4)    Figure 5 and figure 7

Why not combine them?

ANSWER: To make it easier for the readers, we prefer to maintain separate Figures 5 and 7.

(5)    Section 3.4

I see CUP1 interact with CUP2. There is no background why they study their interaction. Is this their initial purpose?

ANSWER: We are sorry, probably we don’t understand this comment. Based on the protein-protein interaction of the CUP1-CUP2 homologs in Arabidopsis, we decided to check if the CUP1-2 proteins of orchids have the same ability. Our results have shown that also in orchids, these proteins can directly interact with each other.

Small suggestions

(1)    Figure 2, figure legend

Please explain color by color, not by word.

ANSWER: We modified this Figure and its legend according to the suggestion.

(2)    P10, L354

“P values*<0,0004”. What is the inner reference gene?

ANSWER: As indicated in the Materials and Methods section, the reference gene is 18S. We added this information in the figure legend.

(3)    P11, L398

There is figure legend of figure 6, but no figure.

ANSWER: We are sorry; in the first submission, we did not include Figure 6. We included Figure 6 in the revised version of the manuscript.

Reviewer 2 Report

The manuscript number: Genes-2067628 entitled “The orchid NAC transcription factors: a focused analysis on the CUP genes” authored by Valoroso et al., is the transcriptomic based study in orchids to understand the role of CUP genes during orchid flower development. The reviewer gone through the manuscript and found that the manuscript is presented as per journal guidelines, the contents of the manuscript is in defined format, table and figures are presented appropriately. The followings are the query and suggestions for improving the quality of the manuscript-

1.      Replace “programs” with “processes” line no. 10

2.      Rewrite the sentence “To check if the CUP homologs .. genes.” For better clarity line no. 15-18

3.      Replace flower formation” with “development” line no. 19

4.      Replace “plant growth” with “plant growth and development” line no. 28

5.      Italicize “NAC” in line no. 31

6.      Check throughout the manuscript for the same carefully, as the name of transcription factor and protein are also may be the same.

7.      Rewrite the sentence “In rare cases, defined as atypical, some .. C-terminal region [21].” For better clarity line no 50-52

8.      Briefly describe the economic value and global cultivation scenario of orchid

9.      Add the geographical location of the experiment site in the method and material section

10.  Figure 1 is ok

11.  The supplementary file (word file having primer details) may be shifted to the manuscript

12.  Method and material section is written and presented in a nice way

13.  Figure 2 to 8 are ok

14.  The results and discussion section is written and presented nicely and the interpretation of results in the discussion is ok

15.  Modify the conclusion section for more soundness.

16.  There are many grammatical mistakes and use of unscientific words which should be removed

Author Response

Dear Editor,

We thank you and the three reviewers for the positive feedback on our MS. We appreciate the comments and suggestions that helped us to improve our work.

The point-by-point answers to reviewers are listed below.

We hope that the revised version of the MS is now suitable for publication in Genes.

Best regards,

Maria Carmen Valoroso,

Francesca Lucibelli

Serena Aceto

REVIEWER 2

The manuscript number: Genes-2067628 entitled “The orchid NAC transcription factors: a focused analysis on the CUP genes” authored by Valoroso et al., is the transcriptomic based study in orchids to understand the role of CUP genes during orchid flower development. The reviewer gone through the manuscript and found that the manuscript is presented as per journal guidelines, the contents of the manuscript is in defined format, table and figures are presented appropriately. The followings are the query and suggestions for improving the quality of the manuscript-

  1. Replace “programs” with “processes” line no. 10

ANSWER: As the “processes” is repeated in the subsequent sentence, we replaced “processes” with “pathways”.

  1. Rewrite the sentence “To check if the CUP homologs .. genes.” For better clarity line no. 15-18

ANSWER: We modified this sentence.

  1. Replace flower formation” with “development” line no. 19

ANSWER: Done.

  1. Replace “plant growth” with “plant growth and development” line no. 28

ANSWER: Done.

  1. Italicize “NAC” in line no. 31

ANSWER: Done.

  1. Check throughout the manuscript for the same carefully, as the name of transcription factor and protein are also may be the same.

ANSWER: We checked throughout the manuscript and corrected the names.

  1. Rewrite the sentence “In rare cases, defined as atypical, some .. C-terminal region [21].” For better clarity line no 50-52

ANSWER: We modified this sentence.

  1. Briefly describe the economic value and global cultivation scenario of orchid

ANSWER: We added short sentences on the economic relevance of orchids in the Conclusion section.

  1. Add the geographical location of the experiment site in the method and material section

ANSWER: Done.

  1. Figure 1 is ok

  1. The supplementary file (word file having primer details) may be shifted to the manuscript

ANSWER: We shifted the supplementary table to the main text (Table 1) and modified the supplementary file accordingly.

  1. Method and material section is written and presented in a nice way

  1. Figure 2 to 8 are ok

ANSWER: We are sorry for missing Figure 6 in the first submission of the manuscript. We added it in the revised version.

  1. The results and discussion section is written and presented nicely and the interpretation of results in the discussion is ok

  1. Modify the conclusion section for more soundness.

ANSWER: We modified the Conclusion section, adding some sentences on the relevance of the studies of orchid flower development.

  1. There are many grammatical mistakes and use of unscientific words which should be removed

ANSWER: We carefully checked the English language and corrected many mistakes.

Reviewer 3 Report

The manuscript entitled “The orchid NAC transcription factors: a focused analysis on the CUP genes” studies transcriptome and genome-wide analysis of orchid NACs, focusing on the NAM subfamily and CUP genes. Orchids especially terrestrial species like Orchis militaris are among the endangered species with little molecular information on their transcriptional basis of flower development. With other comments:

1- However, the manuscript is well-written but, in some parts, it needs a moderate structural and grammatical language revision. Here are some examples:

L28: Among the more thanAmong more than

L29: a large amounta large number

L53: analyzehave analyzed

Please use italic letters for all gene names

2- The title could be changed as you have already done the bioinformatic analysis on this gene. Moreover, please write the full form of abbreviated words in title and abstract (e.g., NAC, NUM, CUP, …..).

3- Introduction is monotone and could be furnished with more information on the studied orchid species.

4- Please specify in material and method the exact origin of the studied material and how propagation materials have been obtained. What is the exact origin of collected O. italica?

5- One of my main questions is the role of O. italica in your study, there is no provided expression or bioinformatic analysis for this species. Can you clarify what you exactly do on this species?

6- Please also provide the cycle and temperature used for Real-Time analysis.

Altogether I rate the manuscript as interesting and it needs minor revision and it could be considered for further evaluation after performing the required revisions

Author Response

Dear Editor,

We thank you and the three reviewers for the positive feedback on our MS. We appreciate the comments and suggestions that helped us to improve our work.

The point-by-point answers to reviewers are listed below.

We hope that the revised version of the MS is now suitable for publication in Genes.

Best regards,

Maria Carmen Valoroso,

Francesca Lucibelli

Serena Aceto

REVIEWER 3

The manuscript entitled “The orchid NAC transcription factors: a focused analysis on the CUP genes” studies transcriptome and genome-wide analysis of orchid NACs, focusing on the NAM subfamily and CUP genes. Orchids especially terrestrial species like Orchis militaris are among the endangered species with little molecular information on their transcriptional basis of flower development. With other comments:

1- However, the manuscript is well-written but, in some parts, it needs a moderate structural and grammatical language revision. Here are some examples:

L28: Among the more than→ Among more than

ANSWER: Done.

L29: a large amount→ a large number

ANSWER: Done.

L53: analyze→ have analyzed

ANSWER: Done.

Please use italic letters for all gene names

ANSWER: We checked and corrected the manuscript.

2- The title could be changed as you have already done the bioinformatic analysis on this gene. Moreover, please write the full form of abbreviated words in title and abstract (e.g., NAC, NUM, CUP, …..).

ANSWER: We added the full name of CUP in the title (CUPULIFORMIS) and added the full name of NAC, NAM and CUP in the abstract.

3- Introduction is monotone and could be furnished with more information on the studied orchid species.

ANSWER: We added some information on P. aphrodite.

4- Please specify in material and method the exact origin of the studied material and how propagation materials have been obtained. What is the exact origin of collected O. italica?

ANSWER: We specified the collection site of O. italica (Sassano, Salerno, Italy). It is challenging the ex situ propagation of O. italica. We do not propagate this material. Our botanist colleagues collect new plants every year during spring.

5- One of my main questions is the role of O. italica in your study, there is no provided expression or bioinformatic analysis for this species. Can you clarify what you exactly do on this species?

ANSWER: To obtain information about the genomic organization of the highest number of orchid CUP genes,  we included Orchis italica in our analysis. The genome is unavailable for this species; however, we have access to samples of O. italica that are part of the orchid collection of the Department of Biology (University of Naples Federico II). We extracted genomic DNA from leaves of O. italica, PCR amplified and sequenced the CUP genes, thus reconstructing their exon/intron structure. We added a sentence about this in the Materials and Methods section. In addition, to avoid confusion, we modified Figure 1 including only the flower of P. aphrodite.

6- Please also provide the cycle and temperature used for Real-Time analysis.

ANSWER: We provided this information in the Materials and Methods section.

Altogether I rate the manuscript as interesting and it needs minor revision and it could be considered for further evaluation after performing the required revisions
